# *Fishy*: Layerwise Fisher Approximation
# for Higher-order Neural Network Optimization

**Abel L. Peirson**[*]
Google Brain & Stanford University
`alpv95@stanford.edu`

**Ehsan Amid**[*]
Google Brain
`eamid@google.com`

**Yatong Chen**
Google Brain & UC Santa Cruz
`ychen592@ucsc.edu`

**Vlad Feinberg**
Google Brain
`vladf@google.com`

**Manfred Warmuth**
Google Research
`manfred@google.com`

**Rohan Anil**
Google Brain
`rohananil@google.com`

## Abstract

We introduce *Fishy*, a local approximation of the Fisher information matrix at each layer for natural gradient descent training of deep neural networks. The true Fisher approximation for deep networks involves sampling labels from the model's predictive distribution at the output layer and performing a full backward pass – *Fishy* defines a Bregman exponential family distribution at each layer, performing the sampling locally. Local sampling allows for model parallelism when forming the preconditioner, removing the need for the extra backward pass. We demonstrate our approach through the Shampoo optimizer, replacing its preconditioner gradients with our locally sampled gradients. Our training results on deep autoencoder and VGG16 image classification models indicate the efficacy of our construction.

## 1 Introduction

Natural gradient descent (NGD) [1] is a second-order update rule that preconditions the gradient direction with the inverse of the model's Fisher information matrix (FIM). NGD corresponds to the steepest descent direction in the Riemannian space associated with the FIM and is equivariant to any differentiable reparameterization of the model weights. However, this property only holds in the small step limit and whether the benefits extend to finite steps remains an open question [16]. Nevertheless, NGD has proven to be remarkably effective in training deep neural networks, and many standard optimizers in deep learning can be seen as approximations of NGD [17, 11].

NGD is challenging to implement for training deep neural networks. Firstly, the FIM requires samples from the model's predictive distribution, with an additional backward pass to calculate the gradients. Secondly, NGD requires calculating the inverse FIM of the whole network, which immediately becomes infeasible even for medium-sized models. K-FAC [17] uses a block diagonal Kronecker factor approximation to address the latter obstacle, considering the FIM for each layer as a Kronecker product of two matrices which can more easily be stored and inverted. However, the Kronecker approximation used in K-FAC becomes challenging to generalize for layers other than fully-connected layers. Moreover, the sampling and additional backward passes keep K-FAC expensive. The Shampoo optimizer [8, 5] approximates the full-matrix AdaGrad preconditioner using the same minibatch gradients of the training examples. Shampoo preconditioning avoids additional sampling or extra backward passes but is no longer related to true Fisher [14, 16] and, as we show experimentally, is outperformed when replaced with a true Fisher approximation. Using a Shampoo approximation of the true Fisher is related to [19]; primary differences are the choice of exponents and

---

[*]Equal Contribution

Has it Trained Yet? Workshop at the Conference on Neural Information Processing Systems (NeurIPS 2022).

dampening term used for per-dimension preconditioners. Our work follows Shampoo's Kronecker product approximation of the preconditioner, which has the advantage of being agnostic to the type of layers. Sun and Nielsen [22] recently proposed a layerwise relative FIM, but do not consider the full form of layers' predictive distributions nor demonstrate scaling to practical architectures.

This paper presents a new layerwise FIM approximation approach, extending the popular Shampoo optimizer to NGD. First, we update Shampoo to an approximate NGD method, sampling from the output layer distribution to construct the preconditioners. Second, inspired by the matching loss [9, 4] construction for LocoProp and layerwise representation learning [3, 2], we define Bregman exponential family distributions for the outputs of each layer based on their activation functions. Sampling locally from these distributions allows preconditioner calculation to be performed in parallel across layers. Third, we compare our approach to standard Shampoo and its Fisher counterpart on a benchmark MNIST deep autoencoder problem and CIFAR100 classification with a VGG16 [20].

## 2 Methodology

**Fisher Information Matrix and Natural Gradient Descent**  Given a probabilistic model $P(\boldsymbol{x}, \boldsymbol{y}|\boldsymbol{\theta})$ with input random variable $\boldsymbol{x}$ and target random variable $\boldsymbol{y}$, the Fisher Information Metric (FIM) is defined in terms of a local Riemannian metric as an approximation to the KL divergence [18],

$$
D_{\mathrm{KL}}\big(P(\boldsymbol{x}, \boldsymbol{y}|\boldsymbol{\theta}), P(\boldsymbol{x}, \boldsymbol{y}|\boldsymbol{\theta} + \mathrm{d}\boldsymbol{\theta})\big) = \int_{\boldsymbol{x}, \boldsymbol{y}} p(\boldsymbol{x}, \boldsymbol{y}|\boldsymbol{\theta}) \log \frac{p(\boldsymbol{x}, \boldsymbol{y}|\boldsymbol{\theta})}{p(\boldsymbol{x}, \boldsymbol{y}|\boldsymbol{\theta} + \mathrm{d}\boldsymbol{\theta})} \, \mathrm{d}\boldsymbol{x} \, \mathrm{d}\boldsymbol{y}
$$

$$
\approx {}^{1}\!/_{2}\, \mathrm{d}\boldsymbol{\theta}^{\top} \underbrace{\mathbb{E}_{\boldsymbol{x}, \boldsymbol{y}|\boldsymbol{\theta}}[\nabla_{\boldsymbol{\theta}} \log p(\boldsymbol{x}, \boldsymbol{y}|\boldsymbol{\theta}) \nabla_{\boldsymbol{\theta}} \log p(\boldsymbol{x}, \boldsymbol{y}|\boldsymbol{\theta})^{\top}]}_{:= \boldsymbol{F}(\boldsymbol{\theta}) \ \text{Fisher Information matrix}} \mathrm{d}\boldsymbol{\theta} \, ,
$$

where the expectation is with respect to the model distribution $P(\boldsymbol{x}, \boldsymbol{y}|\boldsymbol{\theta})$. The result is identical when flipping the order of the arguments in the KL divergence. The FIM corresponds to the Riemannian metric that locally explains the model in the parameter space [15]. This probabilistic model approximates an underlying joint data distribution $P_{\mathrm{d}}(\boldsymbol{x}, \boldsymbol{y})$ from which the data is sampled. Thus, training the model corresponds to finding optimal parameters $\boldsymbol{\theta}^{*}$ that minimize the KL divergence between the data and model distributions. For a discriminative model that generates $P(\boldsymbol{y}|\boldsymbol{x}, \boldsymbol{\theta})$, the marginal distribution $P(\boldsymbol{x}|\boldsymbol{\theta})$ is assumed to be the same as the underlying input data distribution $P_{\mathrm{d}}(\boldsymbol{x})$. Thus, the loss can be written as

$$
L(\boldsymbol{\theta}) = \int_{\boldsymbol{x}, \boldsymbol{y}} p_{\mathrm{d}}(\boldsymbol{x}, \boldsymbol{y}) \log \frac{p_{\mathrm{d}}(\boldsymbol{x}, \boldsymbol{y})}{p(\boldsymbol{x}, \boldsymbol{y}|\boldsymbol{\theta})} \, \mathrm{d}\boldsymbol{x} \, \mathrm{d}\boldsymbol{y} = \underbrace{\mathbb{E}_{\boldsymbol{x}, \boldsymbol{y}}[\log p_{\mathrm{d}}(\boldsymbol{y}|\boldsymbol{x})]}_{\text{constant}} - \mathbb{E}_{\boldsymbol{x}, \boldsymbol{y}}[\log p(\boldsymbol{y}|\boldsymbol{x}, \boldsymbol{\theta})] \, , \quad (1)
$$

where the expectations are applied with respect to $P_{\mathrm{d}}(\boldsymbol{x}, \boldsymbol{y})$. In practice, the empirical loss is formed by a Monte Carlo approximation of the expectation,

$$
L(\boldsymbol{\theta}|\mathcal{X}) = -\frac{1}{n} \sum_{i} \log P(\boldsymbol{y}^{(i)}|\boldsymbol{x}^{(i)}, \boldsymbol{\theta}) \, ,
$$

using training examples $\mathcal{X} = \{(\boldsymbol{x}^{(i)}, \boldsymbol{y}^{(i)})\}_{i=1}^{n}$ that are sampled from $P_{\mathrm{d}}(\boldsymbol{x}, \boldsymbol{y})$.

Given the loss $L(\boldsymbol{\theta}|\mathcal{X})$ on the batch of examples $\mathcal{X}$, the *natural gradient* step [1] is defined as the minimizer of the following linearized objective,

$$
\mathrm{d}\boldsymbol{\theta}_{\mathrm{NGD}} = \arg \min_{\mathrm{d}\boldsymbol{\theta}} \{L(\boldsymbol{\theta}|\mathcal{X}) + \mathrm{d}\boldsymbol{\theta}^{\top} \nabla L(\boldsymbol{\theta}|\mathcal{X}) + {}^{1}\!/_{2\gamma}\, \mathrm{d}\boldsymbol{\theta}^{\top} \boldsymbol{F}(\boldsymbol{\theta}) \, \mathrm{d}\boldsymbol{\theta}\}
$$

$$
= -\gamma \boldsymbol{F}(\boldsymbol{\theta})^{-1} \nabla L(\boldsymbol{\theta}|\mathcal{X}) \, , \quad (2)
$$

in which the loss at $L(\boldsymbol{\theta} + \mathrm{d}\boldsymbol{\theta}|\mathcal{X})$ is approximated by its first-order Taylor expansion. The non-negative multiplier $\gamma > 0$ in the regularizer term controls the step size. Specifically, Equation (2) corresponds to minimizing a linear approximation of the loss at the current parameter $\boldsymbol{\theta}$ while the amount of deviation is measured in terms of the local Mahalanobis distance induced by the FIM.

**Fishy: Layerwise Fisher Construction**  Let $\hat{a}_{\ell}$ denote the pre-activation at layer $\ell \in [L]$ of an $L$-layer network. For a fully-connected layer with weights $\boldsymbol{W}_{\ell}$, we can write $\hat{a}_{\ell} = \boldsymbol{W}_{\ell}\, \hat{\boldsymbol{y}}_{\ell-1}$ and

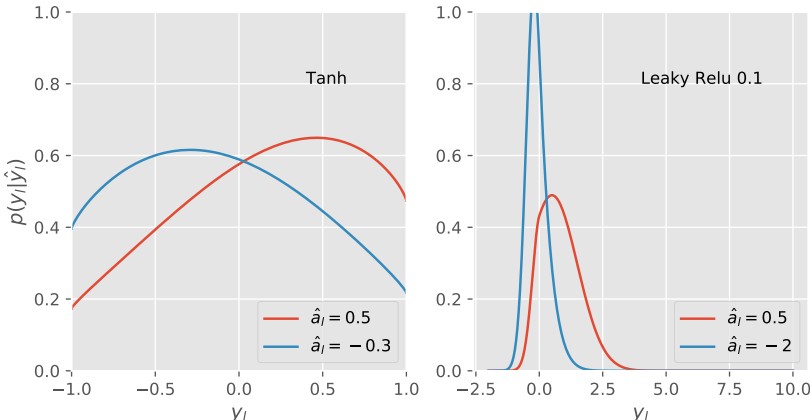

Figure 1: Example of layerwise predictive distributions for tanh (left) and leaky ReLU (right) activation functions.

the post-activation $\hat{\boldsymbol{y}}_\ell = f_\ell(\hat{\boldsymbol{a}}_\ell)$. Here, $f_\ell$ denotes the (elementwise) *strictly-increasing* activation function in layer $\ell \in [L]$. (Note that using this notation, $\hat{\boldsymbol{y}}_0 = \boldsymbol{x}$.) Fishy aims to approximate the Fisher $\boldsymbol{F}$ for each layer *independently* by defining a predictive distribution for $\boldsymbol{y}_\ell$, the layer's output random variable, as

$$p_\ell(\boldsymbol{y}_\ell|\hat{\boldsymbol{y}}_\ell) = \frac{\exp(-D_{F_\ell^*}(\boldsymbol{y}_\ell, \hat{\boldsymbol{y}}_\ell))}{Z(\hat{\boldsymbol{y}}_\ell)} , \qquad (3)$$

where $\hat{\boldsymbol{y}}_\ell$ is the post-activation of the layer given the input $\hat{\boldsymbol{y}}_{\ell-1}$. The term $Z(\hat{\boldsymbol{y}}_\ell)$ is the normalizing partition function that ensures the distribution sums to one. The distribution in Equation (3) belongs to a Bregman exponential family [6] in which the Bregman divergence is induced by the convex function $F_\ell^*$. We apply the layerwise loss construction in [3] to obtain $D_{F_\ell^*}$, which corresponds to the matching loss [9, 13, 4] of the activation function $f_\ell$ induced by its integral function. Intuitively, $D_{F_\ell^*}$ encodes the layer's local geometry; thus, correcting for the gradient direction using a local Fisher approximation can stabilize training and improve convergence [22]. For the linear, sigmoid, and softmax activations, Equation (3) yields the standard Gaussian, Bernoulli, and categorical distributions, respectively. Figure 1 gives some example distributions for tanh and leaky ReLU activations.

To calculate the layer's FIM, we need to calculate the gradient of the *score* function $\log p_\ell(\boldsymbol{y}_\ell|\hat{\boldsymbol{y}}_\ell)$. Taking the gradient with respect to the pre-activation $\hat{\boldsymbol{a}}_\ell$, we have

$$\nabla_{\hat{\boldsymbol{a}}_\ell} \log p_\ell(\boldsymbol{y}_\ell|\hat{\boldsymbol{y}}_\ell) = \boldsymbol{y}_\ell - \hat{\boldsymbol{y}}_\ell - \nabla_{\boldsymbol{a}_\ell} \log Z(\boldsymbol{\theta}) = \boldsymbol{y}_\ell - \hat{\boldsymbol{y}}_\ell - \mathop{\mathbb{E}}_{P_\ell(\boldsymbol{y}_\ell|\hat{\boldsymbol{y}}_\ell)} [\boldsymbol{y}_\ell - \hat{\boldsymbol{y}}_\ell] . \qquad (4)$$

The last term in Equation (4) is non-zero in general for asymmetric distributions induced by activation functions such as tanh and leaky ReLU (see Figure 1). Nevertheless, in our approximation, we discard the last term in Equation (4) to form an upper bound of the local FIM approximation. For a fully-connected layer with $\boldsymbol{\theta}_\ell = \mathrm{vec}(\boldsymbol{W}_\ell)$, we can write

$$\boldsymbol{F}(\boldsymbol{\theta}_\ell) \preccurlyeq \mathop{\mathbb{E}}_{P(\boldsymbol{x}, \boldsymbol{y}_\ell)} \left[ \left( (\boldsymbol{y}_\ell - \hat{\boldsymbol{y}}_\ell) \otimes \hat{\boldsymbol{y}}_{\ell-1} \right) \left( (\boldsymbol{y}_\ell - \hat{\boldsymbol{y}}_\ell) \otimes \hat{\boldsymbol{y}}_{\ell-1} \right)^T \right] , \qquad (5)$$

where $P(\boldsymbol{x}, \boldsymbol{y}_\ell) = P_\mathrm{d}(\boldsymbol{x}) P_\ell(\boldsymbol{y}_\ell|\hat{\boldsymbol{y}}_\ell(\boldsymbol{x}|\boldsymbol{\theta}_\ell))$. The FIM approximation in Equation (5) corresponds to a block-diagonal approximation of the full FIM as the layers are treated independently. This is a common procedure for calculating FIM for large neural networks [17, 22]. With Fishy, we have the flexibility of sampling at every layer or dividing the network into multiple disjoint sections and performing the sampling at the output layer of each section. The sampled gradients are then backpropagated through the corresponding section. In the extreme case where the entire network is treated as a single section, Equation (5) reduces to a block diagonal approximation of the true Fisher by sampling from the model's output predictive distribution [17]. Next, we discuss using the Shampoo approximation to apply the inverse of Equation (5) as the preconditioner at each layer.

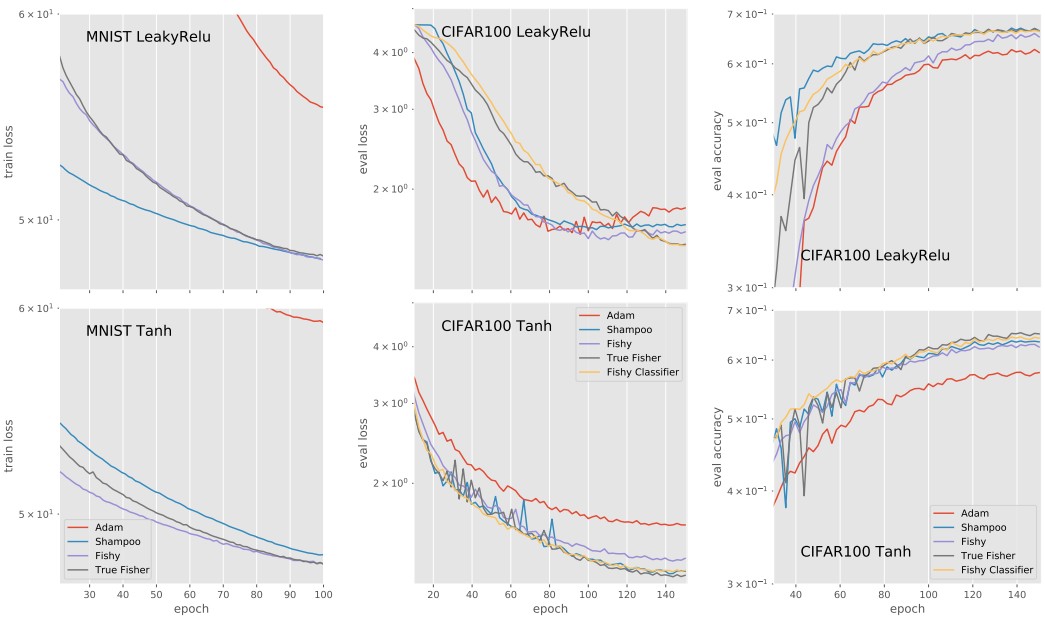

Figure 2: Experimental results on deep autoencoder for MNIST (left column) and VGG16 convolutional network on CIFAR100 for image classification (middle and right columns).

**Fishy via Shampoo Approximation**   For a fully-connected layer with gradient $\boldsymbol{G} \in \mathbb{R}^{d_i \times d_o}$, Shampoo approximates the outer-product of the vectorized gradients $\boldsymbol{g} = \text{vec}(\boldsymbol{G}) \in \mathbb{R}^{d_i d_o}$ as a Kronecker product of two factors

$$\varepsilon \, \boldsymbol{I}_{d_i d_o} + \frac{1}{r} \, \boldsymbol{g}\boldsymbol{g}^\top \preccurlyeq \boldsymbol{L}^{\frac{1}{2}} \otimes \boldsymbol{R}^{\frac{1}{2}} = (\epsilon \boldsymbol{I}_{d_i} + \boldsymbol{G}\boldsymbol{G}^\top)^{\frac{1}{2}} \otimes (\epsilon \boldsymbol{I}_{d_o} + \boldsymbol{G}^\top \boldsymbol{G})^{\frac{1}{2}} \, , \qquad (6)$$

where $r$ is an upper-bound on the rank of the gradients (see [8] for further details), and $\boldsymbol{L}$ and $\boldsymbol{R}$ correspond to the left and right preconditioners. The construction in Equation (6) is more general and can be applied to many types of layers. However, notice that $\boldsymbol{g}$ in Equation (6) corresponds to the average gradient of a batch (and not the per example gradient).

To approximate Equation (5) via Shampoo, we first sample $\boldsymbol{y}_\ell^{(i)}$ from the predictive distribution in Equation (3) given the input sample $\boldsymbol{x}^{(i)}$ in the batch $\{\boldsymbol{x}^{(i)}\}_{i=1}^n$ and the corresponding post-activations $\hat{\boldsymbol{y}}_{\ell-1}$ and $\hat{\boldsymbol{y}}_\ell$ at layer $\ell - 1$ and $\ell$, respectively. Next, instead of the per example sampled gradient $\boldsymbol{g}_s^{(i)} = (\boldsymbol{y}_\ell^{(i)} - \hat{\boldsymbol{y}}_\ell^{(i)}) \otimes \hat{\boldsymbol{y}}_{\ell-1}^{(i)}$, we pass the average sampled gradient $\boldsymbol{g}_s = \mathbf{1}/n \sum_i \boldsymbol{g}_s^{(i)}$ into Shampoo to form the preconditioners. A similar averaging is performed in K-FAC [17] on the sampled gradients before forming the preconditioners. Shampoo then applies the (EMA of the) left $\boldsymbol{L}_s$ and right $\boldsymbol{R}_s$ Fishy preconditioners on the gradient of the training examples $\boldsymbol{G}$,

$$\boldsymbol{G}_{\text{Fishy-NGD}} = \boldsymbol{L}_s^{-\frac{1}{2}} \, \boldsymbol{G} \, \boldsymbol{R}_s^{-\frac{1}{2}} \, . \qquad (7)$$

Notice that the exponents in Equation (7) are different than the original Shampoo formulation $(-1/4)$ as we are interested in the inverse of the local FIM (and not its inverse square root). In practice, the exponent in Shampoo is treated as a tunable hyperparameter.

## 3   Experiments

We perform experiments on two architectures, and datasets, a deep MNIST autoencoder [10] with fully connected hidden layers [1000, 500 × 8, 250, 30, 250, 500 × 8, 1000], a standard benchmark for second-order methods, and a VGG16 CNN architecture [20] applied to CIFAR100. We compare Adam [12], standard Shampoo [5], FIM Shampoo (true Fisher), and our method Fishy. For VGG16, we include a variant of Fishy that only approximates the local FIM for the final fully-connected

layers (called Fishy classifier), letting earlier convolutional layers inherit gradients from the first fully-connected layer's predictive distribution. Hyperparameters for each method are tuned over 400 trials with the Vizier Bayesian optimization toolbox [21, 7], and the trial with the best final objective value is displayed. All Shampoo and Fishy variants share the same tunable hyperparameter ranges. We find significant improvements over standard Shampoo with Fishy or its variants on most of the tasks. Fishy even outperforms FIM Shampoo on some problems. FIM Shampoo can be considered Fishy applied to the output layer only. We plan to open-source the code to reproduce these experiments.

## 4   Conclusion

We introduce Fishy, a layerwise Fisher construction method that allows model parallelism for approximating the FIM. We apply our construction to Shampoo, which only requires simple adjustments but can improve performance on several problems. In the future, we would like to perform more extensive experiments comparing K-FAC, derive a full expectation version of Fishy without requiring sampling, and develop Fishy for the K-FAC approximation. Further treatment of layerwise predictive distributions on their own can also yield interesting research directions.

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
