# OpenReview forum: "Fishy: Layerwise Fisher Approximation for Higher-order Neural Network Optimization"
_NeurIPS.cc/2022/Workshop/HITY — HITY Workshop NeurIPS 2022_

### Official Review · Reviewer_eg7o · 2022-10-17

**Rating:** 1
**Confidence:** 4

**Review:**

This paper develops a new, more efficient way to approximate the Fisher using a layer-wise local approximation, for use with natural gradient descent in deep learning. The manuscript is well written and easy to understand, and provides compelling results on the efficacy of the proposed method on certain problems when used in conjunction with the Shampoo optimizer.

---

### Official Review · Reviewer_prqa · 2022-10-17
**Accept: Interesting proposal of a layer-wise approximation of a neural network’s Fisher information matrix**

**Rating:** 1
**Confidence:** 3

**Review:**

The paper proposes a layer-wise approximation to a neural network’s Fisher information matrix, to avoid multiple costly backward passes, which are usually required to compute (an approximation of) the Fisher.

I recommend to accept the paper, since it enables new and more efficient variations of existing approximate natural gradient descent methods.

Some suggestions to improve the paper:
- Section 1 (Introduction), line 31, footnote: If I’m not missing something, Section 2 does not mention the differences to [17] anymore.
- It would be interesting to hear some intuition on the effect of discarding the last term in Equation (4).
- It would interesting to also show how Fishy can be applied to K-FAC and how the resulting method is different from Fishy + Shampoo.
- The computational complexity of Fishy and regular Shampoo/K-FAC should be compared. Additionally, empirical wall-clock time measurements would be nice.
- Regarding the experiments, you state that “We find significant improvements over Shampoo with Fishy, especially for the tanh activations”. However, this only seems to hold for the train loss in MNIST with tanh activations experiment. Also, it is interesting that True Fisher does not seem to consistently improve convergence/final evaluation metrics, which invites further investigation. Finally, it probably makes sense to tune the hyperparameters of Shampoo and Fishy separately (cf. line 118).

---

### Decision · Program_Chairs · 2022-10-20

Accept